# InfoMAE: Pairing-Efficient Cross-Modal Alignment with Informational Masked Autoencoders for IoT Signals

## Abstract

Standard multimodal self-supervised learning (SSL) algorithms regard cross-modal synchronization as implicit supervisory labels during pretraining, thus posing high requirements on the scale and quality of multimodal samples. These constraints significantly limit the performance of sensing intelligence in IoT applications, where the heterogeneity and the non-interpretability of time-series signals result in abundant unimodal data but scarce high-quality multimodal pairs. This paper proposes InfoMAE, a cross-modal alignment framework that tackles the challenge of multimodal pair efficiency under the SSL setting by facilitating efficient cross-modal alignment of pretrained unimodal representations. InfoMAE achieves *efficient cross-modal alignment* with *limited data pairs* through a novel information theory-inspired formulation that simultaneously addresses distribution-level and instance-level alignment. Extensive experiments on two real-world IoT applications are performed to evaluate InfoMAE's pairing efficiency to bridge pretrained unimodal models into a cohesive joint multimodal model. InfoMAE enhances downstream multimodal tasks by over 60% with significantly improved multimodal pairing efficiency. It also improves unimodal task accuracy by an average of 22% [1].

### ACM Reference Format:

Anonymous Author(s). 2024. InfoMAE: Pairing-Efficient Cross-Modal Alignment with Informational Masked Autoencoders for IoT Signals. In . ACM, New York, NY, USA, 16 pages. https://doi.org/10.1145/nnnnnnn.nnnnnnn

## 1 Introduction

Multimodal Self-Supervised Learning (SSL) algorithms, although achieving unprecedented performance in extensive sensing applications [11, 12, 40, 52, 53], present unique data challenges rarely encountered with unimodal SSL or vision-language domains due to the complexity in acquiring high-quality multimodal pair for IoT signals. The inherent properties of sensory data common in sensing applications result in abundant unimodal signals but scarce multimodal pairs. First, sensory modalities have heterogeneous properties, such as sampling rate, timestamp, or duration, that increase the likelihood of capturing asynchronous events. Consequently, standard IoT multimodal datasets require manual calibration to reduce temporal misalignments or to synchronize between the time-series signals. [40, 57, 62]. Second, raw IoT signals often lack intuitive

[1]We promise to release the source code upon the paper's acceptance.

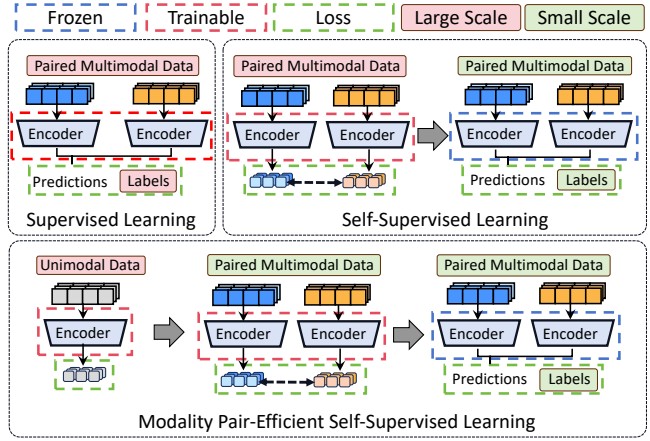

**Figure 1: Comparison of supervised learning, self-supervised learning, and pair-efficient self-supervised learning.**

interpretability. Unlike images or text, where visual features can be easily matched to textual captions, capturing useful signatures between sensing modalities like vibration or frequency waves is challenging. Preprocessing and calibrating these signals requires modality-specific domain knowledge or technical expertise, which is labor-intensive and susceptible to operational errors. Finally, IoT sensors are subject to varying deployment conditions, leading to sparse and noisy data [36]. Each modality can be independently affected by the deployment conditions or environmental factors. For instance, a loud noise source might significantly impact acoustic sensors while having minimal effect on seismic signals. This heterogeneity often results in poor-quality multimodal pairs that are uncorrelated with each other or incomplete datasets with significant gaps and missing data points. These factors contribute to significant challenges in IoT data collection. As IoT networks scale in quantity and the number of modalities, acquiring large-scale, high-quality multimodal pairs becomes increasingly time-consuming, error-prone, and less reliable. The limited multimodal pairs with potential misalignments can introduce uninformative false positive pairs [9, 49], polluting the multimodal feature patterns extracted by the pretrained encoders.

Despite these challenges, most existing multimodal SSL frameworks [1, 35, 48, 56] rely heavily on massive multimodal pairs to learn robust joint representations during the pretraining, but their capability could degrade significantly with insufficient synchronized pairs [44, 72]. On the other hand, independently pretraining each modality on their unimodal data and directly concatenating misaligned modality features for finetuning fails to capture cross-modal interactions that are critical to downstream multimodal tasks [28, 69]. Instead, we observe that with limited multimodal pairs, we can effectively convert independently trained unimodal encoders into a coherent model that sustains strong generalizability

in multimodal tasks. We refer to this process as *pair-efficient SSL*. The relation of pair-efficient SSL for multimodal data compared to standard SSL draws an analogy to the evolvement of SSL compared to supervised learning, as visualized in Figure 1. In supervised learning, manual labels serve as supervision to train encoders for mapping inputs to task-specific labels. Its performance depends heavily on the quantity and quality of human annotations. Self-supervised learning (SSL) mitigates label scarcity by first designating proxy labels from the data properties to learn general semantics with massive unlabeled data, then calibrating the pretrained model to a downstream task with minimal human annotations. Similarly, in multimodal SSL contexts, cross-modal alignment acts as a special form of "supervision", where point-to-point modality correspondence is utilized to identify semantically meaningful and consistent sensory information. Taking another step forward, pair-efficient SSL takes advantage of abundant unimodal data for "independent pretraining", followed by "cross-modal finetuning" with limited multimodal pairs to align unimodal models into a cohesive multimodal model.

In this paper, we propose InfoMAE, a cross-modal learning framework designed to enhance the alignment of unimodal representations using a limited number of multimodal pairs. The key idea behind InfoMAE is to enforce alignment across modalities at both the *distribution* and *instance* levels. Existing contrastive learning frameworks adopt point-to-point alignment to map samples across different modalities to a proximate joint representation [40, 52, 55, 63]. These approaches focus on aligning individual samples, essentially viewing alignment as a local optimization problem that aims to minimize the geometric distances between corresponding samples in the representation space. However, such instance-level approaches face significant challenges with limited multimodal pairs, as they may overfit to the specific pairs available and result in poor generalization with pairing biases. These hinder capturing complex cross-modal relationships, especially when the multimodal pairs are sparse and unevenly distributed. In contrast, InfoMAE takes a more holistic approach by emphasizing *distribution-level* alignment, considering the overall information content of the limited multimodal pairs rather than only focusing on the individual samples. We present a comprehensive analysis of distribution alignment and propose an *information theory-based approach* to formally define the distribution alignment problem in the factorized information space. We formulate this as a differential learning objective to construct (i) shared joint representations as a compact common variable across modalities capable of performing any multimodal task and (ii) private representations holding implicit modality-specific information independent of shared representations. InfoMAE alleviates the strict requirement of exact multimodal sample pairs and can better accommodate potential misalignments in data collection or temporal synchronization, improving the representations learned even with a small-scale multimodal pair.

We extensively evaluate InfoMAE across various combinations of pretrained unimodal domains. InfoMAE achieves exceptional performance gain compared to the standard multimodal SSL paradigm under limited multimodal pairs and outperforms existing works when aligning the unimodal representations. Individual unimodal encoders, in return, can also benefit from the representational structures with improved downstream performance. Additionally, as the number of multimodal pairs scale, InfoMAE also demonstrates versatility as a standard multimodal SSL framework, achieving SOTA performance across real-world IoT applications.

## 2 Analysis of Cross-Modal Alignment

### 2.1 Notation

Consider $M$ sets of unsynchronized modality data $X = \{X_i\}_{i \in M}$, where each set $X_i$ contains unlabeled samples of fixed-length windows partitioned from the time-series signals of the $i$-th modality. Let $N_i = |X_i|$ denote the size of each modality set.

For the $j$-th sample of modality set $i$, we apply Short-Time Fourier Transform (STFT) to obtain its time-frequency representation, $\mathbf{x}_{ij} \in \mathbb{R}^{C_i \times I \times S_i}$, where $C_i$ is the number of input channels, $I$ is the number of time intervals within a sample window, and $S_i$ is the spectrum length in the frequency domain. We have a set of modality encoders $\mathcal{E} = \{E_1, E_2, \ldots, E_M\}$ to extract the modality embeddings of each sample and a set of modality decoders $\mathcal{D} = \{D_1, D_2, \ldots, D_M\}$ to map the samples from the embedding space back to the time-frequency domain $\hat{X} = \{\hat{X}_i\}_{i \in M}$ as a part of the reconstruction process. Additionally, there is a set of multimodal data $X^s = \{X_i^s\}_{i \in M^s}$ consisting of a subset of modalities $M^s \subseteq \hat{M}$, where samples across the modalities are synchronized in time and have equal sizes $|X_1^s| = \cdots = |X_{M^s}^s|$. Note that each synchronized data of modality $i$ can also be a subset of the unsynchronized unimodal, set such that $X_i^s \subseteq X_i$, as any synchronized multimodal data is inherently unsynchronized when considered independently. Finally, we have a set of labeled data for supervised learning and finetuning on a much smaller scale, where each sample has a corresponding label $y_j$ for each downstream task.

### 2.2 Problem Definition

Prior multimodal SSL practices require large-scale, fully synchronized multimodal sets $X^s$ to learn joint multimodal representations that perform well in downstream tasks. However, these approaches often overlook two challenges: (i) *Insufficient multimodal data*: When $|X^s|$ is small, existing methods struggle to learn effective joint representations, and (ii) *Unutilized unimodal data*: The abundance of available unimodal data is often neglected. In IoT applications, the scale of synchronized multimodal sets can be significantly limited due to signal heterogeneities, temporal misalignment, or domain variances, which result in incomplete modalities. This results in more available unimodal data than synchronized multimodal data ($|X_i| \geq |X_i^s|$). However, this abundant unimodal data is excluded from existing multimodal SSL pretraining techniques. To better utilize unimodal data, our problem falls under the SSL setting with unimodal pretrained models and limited multimodal pairs, which consists of two stages:

*Independent Unimodal Pretraining:* For each independent modality data $X_i$, we train a corresponding unimodal encoder $E_i$. The goal is to learn a *holistic unimodal representation* that maximizes downstream unimodal performance after finetuning. Since modality sets $X_i$ are independent, this pretraining is not constrained by the number of synchronized pairs and can, therefore, fully leverage the abundant unimodal data.

*Efficient Cross-Modal Alignment:* Given a set of synchronized modalities data $X^s$ of $M^s \subseteq M$ modalities, we aim to align the

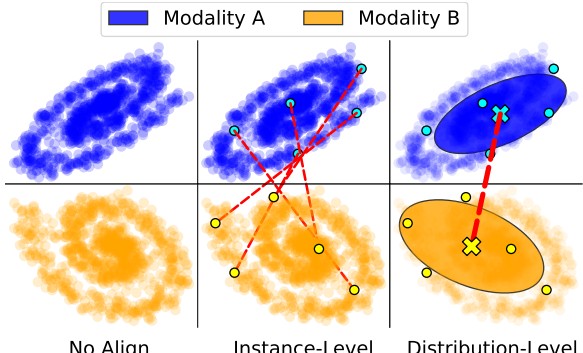

**Figure 2: An illustration of instance-level vs. distribution-level Cross-Modal Alignment**

pretrained encoders efficiently. This alignment projects unimodal representations into joint representations that maximize the downstream multimodal performance after finetuning. The scale of the multimodal alignment should be significantly smaller than the unimodal pretraining $|X_i^s| \lll |X_i|$. In contrast to prior multimodal SSL works focusing on learning robust joint representations on large-scale multimodal data, this work aims to improve the *data efficiency* of learning robust joint representations given only limited multimodal pairs.

## 2.3 Factorization & Distributional Alignment

This section analyzes multimodal representation factorization in the information space and demonstrates how it enables distribution-level alignment of unimodal representations.

*2.3.1 Connection between Factorization and Cross-modal Alignment.* In aligning multimodal representations, prior approaches often rely on contrastive learning to minimize the *modality gap* [39] by pulling representations of different modalities from the same sample closer together while pushing representations from different samples further apart. However, due to the inherent heterogeneity, each modality contains unique, modality-specific information, and enforcing perfect alignment across modalities could potentially hurt the performance in multimodal downstream tasks [29]. To address these challenges, recent works [29, 37, 40] have proposed factorizing modality representations into shared and private subspaces. It preserves both common and modality-specific information and allows for the alignment of shared representations while maintaining independent private representations for downstream tasks. However, these works operate on *instance-level alignment*, and it remains unclear whether this is sufficient when only limited modality pairs are available for learning. The scarcity of paired samples introduces the risk of biased sampling, potentially misleading the alignment process. With this in mind, we analyze a different approach that factorizes the representation in the information space and enforces *distribution-level* alignment to capture a more comprehensive correlation between modalities by *emphasizing their information content rather than just their geometric proximity*. The intuition behind this is that instead of individual sample pairs, we aim to align modalities by the global structure (as shown in Figure 2). When the multimodal

pairs are scarce, the distributional alignment aims to be *resilient to sampling biases* and capture meaningful cross-modal relationships.

*2.3.2 Distributional Alignment through Information-theory based Factorization.* We now formally define the factorization problem in the information space. Without loss of generality, we state the definitions for two modalities, $X = \{X_1, X_2\}$, but they can be generalized to more modalities.

First, we are interested in constructing a compact random variable $U$ (shared representation) that can perform any task that can be achieved using $X_1$ separately and $X_2$ separately. Formally, we define a sufficient common variable as follows.

*Definition 2.1.* (Sufficient Common Variable) $U$ is defined as the sufficient common variable between $X_1, X_2$ if and only if $U = s_1(X_1) = s_2(X_2)$ for some $s_1, s_2$, and

$$(\forall f_1, f_2)\Big(\big[f_1(X_1) = f_2(X_2)\big] \implies \big[(\exists f)f(U) = f_1(X_1) = f_2(X_2)\big]\Big), \tag{1}$$

namely, any common (shared) function between $X_1, X_2$ can be computed using $U$. Building on the sufficient common variable, we define the shared representation to be the most compact form of $U$ with the minimized entropy to ensure that $U$ captures only the essential shared features across modalities.

*Definition 2.2.* (Shared Representation) We refer to a sufficient common variable $U$ with minimal entropy $H(U)$ as the shared representation.

However, it is not clear how to find a sufficient common variable or a shared representation. We show that an approximation of the shared representation can be obtained by solving the following optimization problem, and later in Section 3, we propose the differentiable loss objectives with proof provided in Appendix A.

$$\begin{aligned} \text{minimize} \quad & H(U) \quad \text{s.t. } X_1 \perp\!\!\!\perp X_2 \mid U, \\ & \text{and } (\exists s_1, s_2)\ U = s_1(X_1) = s_2(X_2) \end{aligned} \tag{2}$$

The conditional independence in Equation 2 enforces a form of distributional alignment, ensuring that given the shared representation $U$ is the most compact aligned representation such that $X_1, X_2$ provide no additional information about each other.

Moreover, we define the private representations $V_1, V_2$ between $X_1, X_2$ as follows.

*Definition 2.3.* (Private Representation) $V_1, V_2$ is the private representation of $X_1, X_2$ if they have minimal entropy among the random variables satisfying: $V_1 = p_1(X_1), V_2 = p_2(X_2)$ for some $p_1, p_2$ and there exist functions $g_1, g_2$ such that $X_1 = g_1(V_1, U), X_2 = g_2(V_2, U)$, where $U$ is the shared representation.

Similarly, we look for approximate representations. In particular, we replace equalities with a distance constraint $d$, and independence is replaced by small mutual information. In Section 3, we discuss the detailed implementation of a differentiable loss function to find the approximate representations.

## 3 InfoMAE

This section introduces InfoMAE, a novel cross-modal alignment framework that efficiently aligns unimodal representations at the distribution and instance levels. We provide a detailed overview of InfoMAE's cross-modal alignment module in Figure 3.

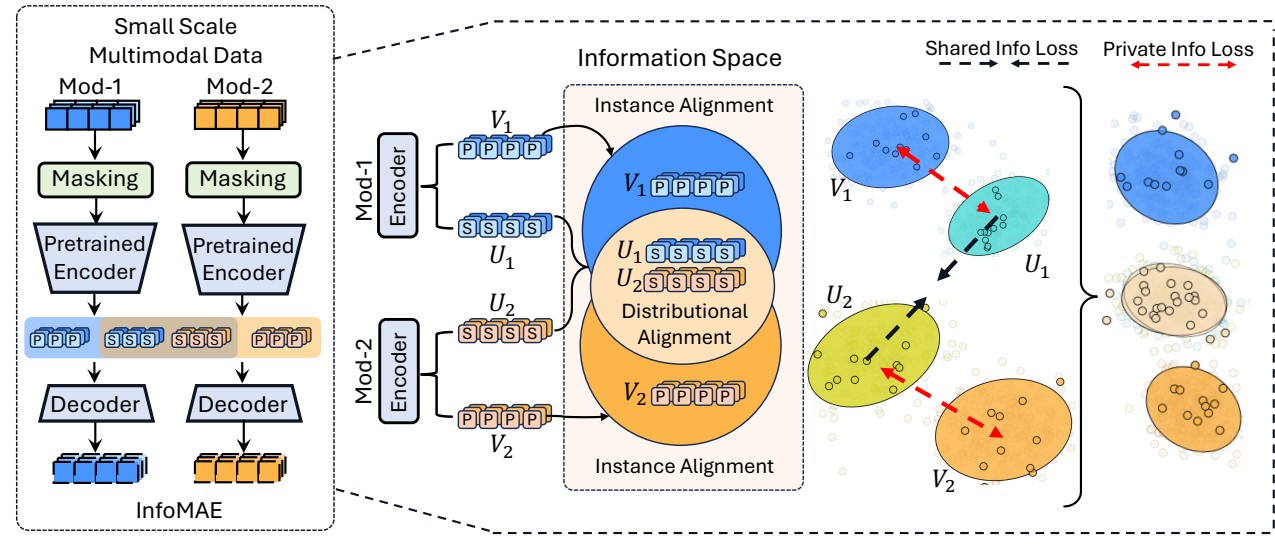

**Figure 3: Overview of InfoMAE's alignment in the information space. InfoMAE adopts an information theory-inspired objective to align the factorized representations. Best viewed in color.**

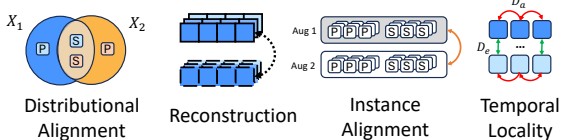

**Figure 4: Key learning objectives of InfoMAE Cross-Modal Alignment.**

## 3.1 Unimodal Pretraining

Unlike standard multimodal SSL that pretrains on synchronized multimodal pairs, we first initiate *unimodal pretraining* on large-scale unsynchronized unimodal data. In the first stage, we pretrain each encoder $E_i$ independently on unimodal data $X_i$ with MAE, which applies mask reconstruction defined as the following for each modality $i \in M$:

$$\mathcal{L}_i^{\text{unimodal}} = ||\hat{X}_i - X_i||^2 \mid \hat{X}_i = D_i(E_i(X_i)). \tag{3}$$

The pretrained unimodal encoders $E_i$ extract a generalized representation for each modality $M_i$. However, they do not guarantee information compatibility between modalities when used together in the downstream tasks. In the following sections, we present InfoMAE's different components (as illustrated in Figure 4) to calibrate the encoders to *explicitly align* the modalities in both the distribution-level and instance-level with only a limited amount of multimodal pair $\mathcal{X}^s$.

## 3.2 Distribution-level Alignment

We begin with the differentiable objective function that we optimize to obtain the (approximate) shared ($U$) and private representations ($V$) defined in Section 2.3.2. To extract $U$ that is a function of both $X_1, X_2$, we equivalently extract $U_1 = F_1^{\text{shared}}(E_1(X_1)), U_2 = F_2^{\text{shared}}(E_2(X_2))$, where $F_1, F_2$ are 2-layer MLP projectors that maps the general representation into shared and private representations, and enforce a constraint that $U_1 = U_2$. Similarly, we extract $V_1 =$

$F_1^{\text{private}}(E_1(X_1)), V_2 = F_2^{\text{private}}(E_2(X_2))$. We use $\mathcal{U} = \{U_1, U_2\}$ and $\mathcal{V} = \{V_1, V_2\}$ for the extracted shared and private representations, respectively.

*3.2.1 Shared Representation.* . As described in Section 2, we aim to find the shared representation $U$ that solves the optimization problem in Definition (2.2). However, due to the difficulty of the optimization problem [2] and the possibility that a shared representation does not exist, we instead approximate the shared representation by minimizing the following objective

$$\mathcal{L}_{\text{info}}^{\text{shared}} = \alpha d(U_1, U_2) + \beta(H(U_1) + H(U_2)) \\ + I(X_1; X_2 \mid U_1) + I(X_1; X_2 \mid U_2), \tag{4}$$

where $\alpha$ and $\beta$ are the hyperparameters controlling the weight of each term, and $d(\cdot)$ is a distance measure. The first two terms in the loss function aim to find $U_1 = U_2$ with minimal entropy, while the last two terms aim to impose conditional independence of $X_1, X_2$ given $U_1$ or $U_2$. We would like to note that the entropy and conditional mutual information listed in Eq. (4) are not easy to compute or differentiate. To alleviate this, we reduce these terms into probabilistic density functions below:

$$\mathcal{L}_{\text{info}}^{\text{shared}} = \alpha d(U_1, U_2) + \sum_{i=1}^{2} \mathbb{E}_{X_1, X_2, U_i} \left[ \log \frac{p_{X_1, X_2, U_i}}{p_{X_1} p_{X_2} p_{U_i}} \right. \\ \left. + (1-\beta) \log \frac{p_{X_i, U_i}}{p_{X_i} p_{U_i}} + \log \frac{p_{X_{3-i}, U_i}}{p_{X_{3-i}} p_{U_i}} \right]. \tag{5}$$

Due to the space limit, we leave the detailed proof and discussion in Appendix A. To further enhance the differentiability of Eq. (5) by avoiding directly computing the probabilistic density ((*e.g.*, $\log \frac{p_{X_1, X_2, U_i}}{p_{X_1} p_{X_2} p_{U_i}}$)), we follow [31, 50, 60] and utilize the *density-ratio trick* to train a discriminator $\mathcal{R}$, which given $X_1, X_2, U$, outputs

---

[2]The optimization problem in Definition (2.2) is non-convex with a possibly infinite number of variables.

the probability that $X_1, X_2, U$ are generated from $p_{X_1,X_2,U_i}$, instead of $p_{X_1} p_{X_2} p_{U_i}$. The density ratio can then be estimated as

$$\log \frac{p_{X_1,X_2,U_1}}{p_{X_1} p_{X_2} p_{U_1}} = \log \frac{\mathcal{R}(X_1;X_2;U_1)}{1 - \mathcal{R}(X_1;X_2;U_1)}. \quad (6)$$

We train the discriminators jointly with the encoders and summarize the training configurations for both in Appendix E.

*3.2.2 Private Representation.* . As the decoders take both the shared and private representations as input, the self-reconstruction objective would enforce the private representations $V$ to capture the implicit modality-specific information. Following Definition 2.3, we minimize the entropy of the private representations $(V_1, V_2)$. In addition, for each modality, we expect the private and shared representations to be independent. To better guide the learning process, we explicitly minimize their mutual information. The objectives of the private representations can be summarized as the following:

$$\mathcal{L}_{\text{info}}^{\text{private}} = \gamma H(V_1) + \gamma H(V_2) + \epsilon I(V_1;U_1) + \epsilon I(V_2;U_2), \quad (7)$$

where $\gamma$ and $\epsilon$ are used as the hyperparameters for private entropy and shared private independence. Similar to Eq.(5), we apply *density-ratio trick* (Eq.(6)) to estimate each term in Eq. (7).

While the formulation effectively aligns modality representations within the information space, it depends on further learning objectives to ensure they are meaningful for downstream tasks. Next, we will describe the additional components of InfoMAE that are designed to capture meaningful representations."

## 3.3 Self Reconstruction

InfoMAE applies simple MAE objective to enforce that the learned representation captures the critical semantical information through reconstruction loss. Following [24], we randomly mask out 75% of the patched input. To ensure both the shared and private representation are meaningful, the decoder takes in the concatenated shared and private representation $\mathbf{h}_{ij} = \mathbf{u}_{ij} || \mathbf{v}_{ij}$ to reconstruct the input $\hat{\mathbf{x}}_{ij}$. We take the MSE loss on the masked portion between the reconstructed $\hat{\mathbf{x}}_{ij}$ and the original input $\mathbf{x}_{ij}$ with $\delta$ as the hyperparameter and $D_i(\cdot)$ as the decoder for $i$-th modality.

$$\mathcal{L}_{\text{reconstruction}} = \delta \sum_{i \in M} \sum_{j \in B} ||\mathbf{x}_{ij} - \hat{\mathbf{x}}_{ij}||^2 \mid \hat{\mathbf{x}}_{ij} = D_i(\mathbf{h}_{ij}). \quad (8)$$

## 3.4 Instance-level Alignment

Augmentations are primarily used to generate different views for private-space contrastive learning in most existing works [29, 37, 40]. However, we argue that the transformation invariance property should be reflected in both private and shared representations to understand the instance variances. Thus, InfoMAE adds a contrastive loss on the concatenated representation of the shared and private spaces $\mathbf{h}_{ij}$ by treating two randomly different augmented views as the positive pairs with $\tau$ as the temperature hyperparameter.

$$\mathcal{L}_{\text{aug}} = -\lambda \sum_{i \in M} \sum_{j \in B}$$
$$\log \frac{\exp\left(\mathbf{h}_{ij} \cdot \mathbf{h'}_{ij}/\tau\right)}{\sum_{k \neq j \in B} \exp\left(\frac{\mathbf{h}_{ij} \cdot \mathbf{h}_{ik}}{\tau}\right) + \sum_{k \in B} \exp\left(\frac{\mathbf{h}_{ij} \cdot \mathbf{h'}_{ik}}{\tau}\right)}. \quad (9)$$

**Table 1: Cross-modal Alignment Dataset. Domains used for supervised evaluation are bolded.**

| Application | Modalities | Domains |
|---|---|---|
| Moving Object Detection (MOD) | seismic acoustic | domain M domain G domain T |
| Human Activity Recognition (HAR) | accelerometer gyroscope magnetometer | PAMAP2 RealWorld-HAR |

## 3.5 Temporal Locality

We apply a simple ranking constraint to learn *temporal locality* of time-series signals. During pretraining, a sequence sampler randomly selects a batch of sequences consisting of a fixed number of consecutive samples, while the samples across sequences are distant in time. We define $C_{ss'}$ as the average Euclidean distance of all sample embedding pairs between the sequence $s$ and $s'$ of length $L$. Then, we define the temporal constraint as:

$$\mathcal{L}_{\text{temporal}} = \eta \sum_{s \in B} \sum_{s' \neq s \in B} \max\left(C_{ss} - C_{ss'} + 1, 0\right)$$
$$C_{ss'} = \sum_{i=1}^{L} \sum_{j=1}^{L} d_{\text{Euclidean}}(s_i, s'_j), \quad (10)$$

where $d_{\text{Euclidean}}$ denotes the Euclidean distance, $C_{ss}$ and $C_{ss'}$ are the average intra-sequence $(D_a)$ and inter-sequence distances $(D_e)$, and the added 1 is the margin indicating the minimum gap between the two distances. $\eta$ is used as the hyperparameter to control the weight of the temporal constraint.

*3.5.1 Overall Cross-Modal Alignment Objectives.* Finally, the overall training objective of InfoMAE for the cross-modal alignment stage can be summarized as follows:

$$\mathcal{L} = \mathcal{L}_{\text{reconstruction}} + \mathcal{L}_{\text{info}}^{\text{shared}} + \mathcal{L}_{\text{info}}^{\text{private}} + \mathcal{L}_{\text{aug}} + \mathcal{L}_{\text{temporal}}. \quad (11)$$

InfoMAE adopts both distribution-level and instance-level alignment of each modality's factorized shared and private representations. Since the cross-modal alignment of InfoMAE is also a generalized multimodal framework, we would also like to note that this objective can be used as the joint multimodal pretraining objective.

## 4 Evaluations

This section evaluates InfoMAE's paired-data efficiency compared to existing multimodal SSL frameworks on cross-modal alignment. We first describe the experimental setup. Then, we examine InfoMAE's finetuning performance after cross-modal alignment with limited multimodal pairs across two real-world applications. We further demonstrate InfoMAE's flexibility and lastly ablate InfoMAE to understand its performance.

### 4.1 Experimental Setup

*4.1.1 Backbone Encoder.* We use SWIN-Transformer (SW-T) [41] as the backbone encoder. SW-T computes local attention within shifting windows on the input patches to reduce time complexity. Implementation details and encoder configurations for each application are listed in Appendix D and E.

**Table 2: Linear Probing Performance of Moving Object Detection on Domain M by aligning pretrained unimodal encoders. $A_{Sei}\|B_{Aco}$ means seismic encoder from domain A and acoustic encoder from domain B are used for alignment.**

| Framework | Aligned Domains | | $T_{Sei} \| M_{Aco}$ | | $G_{Sei} \| T_{Aco}$ | | $T_{Sei} \| T_{Aco}$ | | $G_{Sei} \| M_{Aco}$ | | $T_{Sei} \| G_{Aco}$ | |
| --- | --- | --- | --- | --- | --- | --- | --- | --- | --- | --- | --- | --- |
| | Joint Pretrain | Modal Alignment | Acc | F1 | Acc | F1 | Acc | F1 | Acc | F1 | Acc | F1 |
| Unimodal Concat | ✗ | ✗ | 0.6731 | 0.6699 | 0.5392 | 0.5281 | 0.4454 | 0.4366 | 0.7247 | 0.7217 | 0.6584 | 0.6543 |
| CMC [63] | ✗ | ✓ | 0.6792 | 0.6702 | 0.4313 | 0.4356 | 0.4173 | 0.4032 | 0.6919 | 0.6877 | 0.6497 | 0.6335 |
| FOCAL [40] | ✗ | ✓ | 0.7462 | 0.7432 | 0.6249 | 0.6249 | 0.5613 | 0.5579 | 0.7549 | 0.7527 | 0.7194 | 0.7160 |
| GMC [55] | ✗ | ✓ | 0.7354 | 0.7317 | 0.6591 | 0.6523 | 0.4756 | 0.4720 | 0.8044 | 0.8053 | 0.7247 | 0.7211 |
| SimCLR [6] | ✗ | ✓ | 0.3061 | 0.2742 | 0.2873 | 0.2609 | 0.2974 | 0.2758 | 0.2981 | 0.2698 | 0.2800 | 0.2308 |
| TNC [64] | ✗ | ✓ | 0.1969 | 0.0815 | 0.1788 | 0.1312 | 0.1855 | 0.1021 | 0.1929 | 0.0896 | 0.1949 | 0.1041 |
| TSTCC [15] | ✗ | ✓ | 0.3001 | 0.2706 | 0.2639 | 0.2393 | 0.2867 | 0.2432 | 0.3048 | 0.2842 | 0.2860 | 0.2337 |
| **InfoMAE** | ✗ | ✓ | **0.7950** | **0.7929** | **0.6986** | **0.7007** | **0.5928** | **0.5908** | **0.8326** | **0.8324** | **0.7636** | **0.7537** |
| Joint Pretrain | ✓ | ✗ | Acc: 0.3329 | | | | | | F1: 0.3039 | | | |

*4.1.2 Datasets.* Our experiments focus on two real-world applications: Moving Object Detection (MOD) and Human Activity Recognition (HAR). The MOD application contains vibration-based datasets using seismic and acoustic sensors. The HAR application consists of publicly released IMU sensor datasets (accelerometer, gyroscope, and magnetometer) collected from many human subjects performing various daily activities. To evaluate cross-modal alignment, we simulate a practical scenario where the pretrained domains differ significantly to reflect the diverse signals across different IoT application domains. Under this setting, we have unsynchronized unimodal data from different domains: MOD consists of data from three separately collected domains (M, G, T), each with different targets, terrains, and environmental conditions. HAR consists of data from two publicly released datasets (RealWorld-HAR [62] and PAMAP2 [57]). We pretrain unimodal encoders with only the unimodal data from different domains and then use a limited amount of synchronized multimodal pairs for cross-modal alignment and downstream finetuning. For joint pretraining, we pretrain on the massive available synchronized multimodal pairs. We summarize the data used in Table 1 and describe these applications and domains in more detail in Appendix B.

*4.1.3 Baselines.* We extensively evaluate InfoMAE with different SSL baselines including unimodal contrastive (SimCLR[7], MoCo[8]), multimodal (CMC[63], GMC[55], FOCAL [40]) contrastive, temporal contrastive (TNC[64], TSTCC[15]), and MAE based frameworks (MAE[24], CAV-MAE[18]). We describe these baselines in more detail in Appendix F.

## 4.2 Cross-Modal Alignment Evaluation

*4.2.1 Cross-Modal Alignment on MOD.* We evaluate InfoMAE against prior CL works [7, 15, 40, 55, 63, 64] on cross-modal alignment with various combinations of unimodal models pretrained with different domains. We align the encoders with a small scale of multimodal pairs (5% of the unimodal data scale) and an even smaller subset of labeled multimodal pairs from domain M for finetuning. This application involves two modalities (seismic and acoustic). Therefore we represent the domains of the unimodal representations with two

letters (*e.g.*, $T_{Sei}\|G_{Aco}$ means aligning seismic encoder pretrained on domain T and acoustic encoder pretrained on domain G).

In addition to the prior CL baselines, we also show the performance for direct concatenation of the pretrained unimodal representations without any alignment and for Joint Multimodal Pretraining on the same amount of synchronized multimodal pairs. We present the accuracy and F1-score after finetuning in Table 2, InfoMAE consistently outperforms the unimodal concatenation by a significant margin since direct concatenation fails to exploit cross-modal correspondence. CMC and other unimodal SSL frameworks even have negative impacts compared to direct concatenation, indicating that unimodal objectives or simply aligning the multimodal representations without considering the modality discrepancy could hurt the downstream performance. InfoMAE also achieves better results than FOCAL and GMC, underscoring the benefits of enforcing distribution-level alignment over instance-level alignment in downstream tasks with limited multimodal data. When the same amount of multimodal data is used for Joint Multimodal Pretraining, the significant gap between the aligned unimodal models and the joint pretrained multimodal model suggests the feasibility of transferring pretrained unimodal representations to multimodal representations with only limited (5%) synchronized multimodal data. It is noteworthy that some domain combinations ( e.g., GT, TT, TG) do not even overlap with data from the alignment and finetuning set (MOD).

*4.2.2 Cross-Modal Alignment on HAR.* Besides MOD application, we also evaluate InfoMAE on HAR applications. In contrast to MOD evaluation, which aligns unimodal encoders pretrained on different domains, we analyze how additional unsynchronized data from the same domains could assist the downstream performance given the limited number of multimodal pairs. Here, we independently pretrain all unimodal encoders on unsynchronized IMU data from either PAMAP2, RealWorld-HAR, or Combined, which is the concatenation of the former two. Then, we use a small portion of the synchronized multimodal data pairs from PAMAP2 for cross-modal alignment and downstream finetuning. We present the results in Table 4. InfoMAE consistently achieves the best performance, with an average of 4.09% and 5.16% improvements in accuracy and the

**Table 3: Alignment performance (MM) with different multimodal pair ratios from MOD.**

| Multimodal Data | Supervised | | Joint Pretrain | | CMC | | GMC | | FOCAL | | **InfoMAE** | |
|---|---|---|---|---|---|---|---|---|---|---|---|---|
| | Acc | F1 | Acc | F1 | Acc | F1 | Acc | F1 | Acc | F1 | Acc | F1 |
| 5% | | | 0.3329 | 0.3039 | 0.7087 | 0.6989 | 0.8614 | 0.8616 | 0.8694 | 0.8668 | **0.8828** | **0.8808** |
| 15% | 0.5740 | 0.5663 | 0.6142 | 0.6104 | 0.8111 | 0.8062 | 0.8781 | 0.8753 | 0.8727 | 0.8703 | **0.9049** | **0.9028** |
| 25% | | | 0.7071 | 0.7938 | 0.8433 | 0.8372 | 0.8774 | 0.8759 | 0.8848 | 0.8831 | **0.9290** | **0.9270** |
| 50% | | | 0.8942 | 0.8920 | 0.8754 | 0.8724 | 0.8948 | 0.8938 | 0.9009 | 0.8994 | **0.9377** | **0.9367** |

**Table 4: Linear Probing performance of HAR on PAMAP2 by aligning pretrained unimodal encoders.**

| Unimodal Pretrain Domain | Combined | | PAMAP2 | | RealWorld-HAR | |
|---|---|---|---|---|---|---|
| Multimodal Alignment Domain | PAMAP2 | | PAMAP2 | | PAMAP2 | |
| Metric | Acc | F1 | Acc | F1 | Acc | F1 |
| Concat | 0.7843 | 0.7000 | 0.7763 | 0.6210 | 0.5675 | 0.4187 |
| CMC | 0.7334 | 0.6508 | 0.7285 | 0.6788 | 0.7010 | 0.5956 |
| FOCAL | 0.7922 | 0.7129 | 0.7354 | 0.6327 | 0.7643 | 0.6243 |
| GMC | 0.7314 | 0.5915 | 0.7344 | 0.5869 | 0.7414 | 0.5816 |
| SimCLR | 0.7299 | 0.6190 | 0.7075 | 0.5426 | 0.7225 | 0.5581 |
| TNC | 0.5431 | 0.4080 | 0.5889 | 0.4824 | 0.6378 | 0.5167 |
| TSTCC | 0.7299 | 0.6003 | 0.7065 | 0.5773 | 0.7354 | 0.5864 |
| **InfoMAE** | **0.8261** | **0.7303** | **0.8117** | **0.7175** | **0.7912** | **0.6901** |

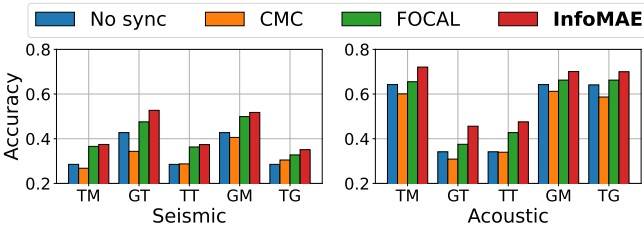

**Figure 5: Unimodal linear probing accuracy of MOD with and without cross-modal alignment.**

F1-score compared to the best-performing baseline, FOCAL. The improvement is most significant in aligning unimodal encoders pretrained on RealWorld-HAR, which completely differs from the alignment set (PAMAP2). This further demonstrates InfoMAE's robustness as an alignment framework with a limited amount of multimodal pairs, reflecting its superior ability to utilize the unimodal data better even when they are from different domains.

## 4.3 Unimodal Evaluation with Cross-Modal Alignment

We analyze how incorporating the multimodal correspondences into each unimodal encoder after alignment could benefit the downstream tasks. Figure 5 shows the accuracy for seismic and acoustic modalities before and after cross-modal alignment in the MOD application. With limited multimodal pairs, the pretrained unimodal encoders could gain the most significant performance improvements with InfoMAE. This emphasizes the InfoMAE's superior efficiency in enforcing cross-modal correspondence to each modality to improve their downstream performance, with only a few multimodal pairs required. With InfoMAE, the aligned unimodal model

**Table 5: Performance of Joint Pretraining on MOD (seismic and acoustic) dataset and then finetuned on unseen domains.**

| Domain | **Domain G** | | **Domain T** | |
|---|---|---|---|---|
| Frameworks | Acc | F1 | Acc | F1 |
| CMC [8] | 0.7924 | 0.7897 | 0.6791 | 0.6776 |
| FOCAL [40] | 0.9137 | 0.9111 | 0.8156 | 0.8130 |
| GMC [55] | 0.7986 | 0.7947 | 0.3457 | 0.3387 |
| MoCo [8] | 0.8719 | 0.8688 | 0.7500 | 0.7483 |
| SimCLR [6] | 0.8418 | 0.8386 | 0.7288 | 0.7207 |
| TNC [64] | 0.6916 | 0.6797 | 0.5680 | 0.5625 |
| TSTCC [15] | 0.7080 | 0.7004 | 0.5804 | 0.5766 |
| MAE [24] | 0.6708 | 0.6642 | 0.4421 | 0.4365 |
| CAV-MAE [18] | 0.5507 | 0.5282 | 0.3457 | 0.3387 |
| **InfoMAE** | **0.9196** | **0.9186** | **0.8546** | **0.8535** |

can generate the most holistic representations through distributional alignment compared to geometric alignment (CMC, FOCAL).

## 4.4 Multimodal Pairing Efficiency

We also evaluate InfoMAE's alignment performance at varying amounts of multimodal data for MOD application in Table 3. We align both encoders pretrained from domain M (MM) and compare them to standard joint pretraining with different ratio of multimodal data. Additionally, we provide supervised training results on the same amount of labeled multimodal data used for finetuning. InfoMAE consistently achieves superior multimodal data efficiency, with minimal degradation as we reduce the number of multimodal data. In general, InfoMAE has an average of 3.42% gain over the highest-performing baselines and over 60% compared to joint model pretraining, which performs poorly in the absence of multimodal data. Note that the joint pretraining even performs worse than the supervised approach with only 5% of multimodal data, indicating the standard self-supervised pretraining fails to learn effective representations with an insufficient amount of synchronized multimodal data. In contrast, the two-stage learning paradigm of InfoMAE leveraging widely available unsynchronized unimodal data could effectively mitigate this problem.

## 4.5 Standard Mutimodal Pretraining on Large-scale Synchronized Dataset

While InfoMAE excels as an efficient cross-modal alignment framework under limited pairs, it also demonstrates remarkable flexibility as a standard multimodal SSL framework. We evaluate InfoMAE

**Table 6: Ablation Results of InfoMAE Cross-Modal Alignment.**

| Frameworks | TM | | GT | | TT | | GM | | TG | |
|---|---|---|---|---|---|---|---|---|---|---|
| | Acc | F1 | Acc | F1 | Acc | F1 | Acc | F1 | Acc | F1 |
| noTemp | 0.6946 | 0.6902 | 0.5881 | 0.5884 | 0.5044 | 0.4888 | 0.7435 | 0.7432 | 0.6651 | 0.6570 |
| noShared | 0.7683 | 0.7595 | 0.6504 | 0.6515 | 0.5298 | 0.5232 | 0.8125 | 0.8116 | 0.7395 | 0.7351 |
| noPrivate | 0.5479 | 0.4732 | 0.4180 | 0.3402 | 0.2873 | 0.1812 | 0.6259 | 0.5519 | 0.5399 | 0.5519 |
| noAug | 0.7863 | 0.7823 | 0.6973 | 0.6967 | 0.5881 | 0.5868 | 0.8232 | 0.8225 | 0.7924 | 0.7879 |
| **InfoMAE** | **0.7950** | **0.7929** | **0.6986** | **0.7007** | **0.5928** | **0.5908** | **0.8326** | **0.8324** | **0.8326** | **0.8324** |

against prior state-of-the-art works on Joint Multimodal Pretraining using abundant multimodal pairs, as shown in Table 5. We use synchronized, unlabeled multimodal data from the MOD dataset to pretrain backbone encoders. Then we freeze the pretrained encoders and perform linear probing using labeled multimodal data from domains $G$ and $T$, as described in Section 4.1. InfoMAE consistently outperforms the MAE-based framework and achieves better performance than other contrastive baselines. We leave more evaluation on Joint Multimodal Pretraining across four real-world datasets to Appendix G. Prior works, primarily designed for joint multimodal pretraining, often struggle with limited multimodal pairs and show significant performance degradation. In contrast, InfoMAE not only improves multimodal pairing efficiency but maintains high performance with minimal performance degradation.

## 4.6 Ablation Studies

Finally, we study how each module of InfoMAE contributes to its performance through ablation studies. We evaluate four variants of InfoMAE by removing temporal, shared, private, and augmentation components in Table 6. The absence of either shared or private components leads to a significant degradation, implying the significance of factorized representation for cross-modal alignment. The drop in performance after removing temporal locality constraints also indicates the importance of learning temporal correspondence for time-series signals. Without temporal locality, the learned representations lose crucial temporal correspondence and can significantly compromise the ability to learn multimodal correspondences on top of the unimodal representations. Conversely, InfoMAE without augmentations does not significantly reduce the performance, demonstrating its robustness toward augmentation choices, in contrast to many contrastive learning frameworks that require careful selection of augmentations to avoid representational collapses.

## 5 Related Works

**Self-Supervised Multimodal Learning.** Self-supervised learning (SSL) techniques, such as Contrastive Learning (CL) and masked reconstructions, have achieved significant success in visual, textual, and time-series representation learning [5, 15, 16, 19, 56, 59, 64, 75, 77, 79]. Masked reconstruction learns informative representations by reconstructing masked inputs [4, 13, 24, 34, 74], with various masking strategies explored [2, 30, 78], and extended to time-frequency spectrograms [27, 51] and videos [20, 65]. Multimodal representation learning has become increasingly important with diverse applications [3, 38, 57, 58, 80]. Recent works leverage CL to learn correspondences between modalities [11, 52, 54, 55, 63, 67, 81],

and others pretrain unified encoders for multimodal representations [26, 47]. Factorized Multimodal Learning [25, 29, 37, 40, 68] further decouples multimodal learning by acknowledging both modality-specific and modality-shared information. FOCAL [40] proposed contrastive learning objectives to learn shared and private representation in the orthogonal space. FactorizedCL [37] separates the shared and private space based on their relevance to the downstream tasks. Some works [18, 71] combine CL with MAE to capture cross-modal correspondence. Yet, these works minimize the geometric modality gap to learn cross-modal correspondences and rely on massive amounts of multimodal data for joint multimodal pretraining. In contrast, InfoMAE minimizes the information modality gap to further enhance the downstream performance. In reducing multimodal data pairs for training, many works [45, 66, 70] propose to impute missing modality pairs through feature generations. Wang *et al.* [72] proposes using CL to align multimodal encoders through an anchor modality yet still overlooking unimodal data. In contrast, InfoMAE minimizes the reliance on multimodal data by taking advantage of a large amount of unimodal data.

**Multimodal Information Theory**. There has been a long history of exploring common information between random variables in information theory [17, 73, 76], and it is still an active research field [21–23, 61]. However, it remains challenging to compute the common information in practical applications. Kleinman1 *et al.* [33] combines Variational Autoencoders with Gacs-Korner Common Information. Mai *et al.* [46] proposes to measure the information redundancy for multimodal data. However, they do not explicitly consider the unique information for factorization. InfoMAE adopts the informational factorization considering both private and shared information to construct a joint representation in a task-agnostic manner rather than extracting task-related information like [37].

## 6 Discussion & Conclusion

We proposed InfoMAE, a pairing-efficient multi-stage SSL paradigm for multimodal IoT sensing. It first pretrains independent modality encoders on large-scale unimodal data sets. Then, it leverages a novel information theory-based optimization to achieve distributional cross-modal alignment with only limited multimodal pairs. Extensive evaluations compared to standard multimodal SSL frameworks demonstrated the superior efficiency and effectiveness of InfoMAE across multiple real-world IoT applications. We believe it opens new opportunities for developing more data-efficient and qualitative self-supervised multimodal models. In the Appendix, we provide more implementation details and evaluations.

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

# Appendix

The appendix of this paper is structured as follows.

- A Proof of Information Formulations.
- B Datasets.
- C Data Preprocessing.
- D Backbone Model.
- F Baselines.
- E Training Configurations.
- G Additional Evaluation.

## A Information Formulation

This section provides detailed proof of the proposed information formulation for cross-modal alignment.

### A.1 Proof of the Equivalence between (1) and (2) in Definitions

We first show the equivalence between the condition (1) and the constraints in (2) by proving the following proposition.

PROPOSITION A.1. *For random variables $X_1, X_2$, if $U = s_1(X_1) = s_2(X_2)$, and there exists $W = g_1(X_1) = g_2(X_2)$ such that $X_1 \perp\!\!\!\perp X_2 \mid W$, then the following two statements are equivalent.*

*(a) $(\forall f_1, f_2)\Big([f_1(X_1) = f_2(X_2)] \implies [(\exists f)f(U) = f_1(X_1) = f_2(X_2)]\Big).$*

*(b) There is a one-to-one mapping between $W$ and $U$ (i.e., $X_1 \perp\!\!\!\perp X_2 \mid U$).*

PROOF. We first prove the direction (b) $\implies$ (a) using properties of basic information-theory measures (Chapter 2 in [10]). For any $f_1, f_2$ such that $f_1(X_1) = f_2(X_2)$, we have

$$0 \overset{(i)}{=} I(X_1; X_2|U) \overset{(ii)}{\geq} I(f_1(X_1); f_2(X_2)|U) \overset{(iii)}{\geq} 0, \quad (12)$$

where $(i)$ follows that $X_1$ and $X_2$ are independent conditioned on $U$; $(ii)$ is due to the data processing inequality of mutual information; and $(iii)$ is because the mutual information is always non-negative. (12) implies that $I(f_1(X_1); f_2(X_2)|U) = 0$. In addition, since $I(f_1(X_1); f_2(X_2)|U) = H(f_1(X_1)|U) - H(f_1(X_1)|f_2(X_2), U)$ and $H(f_1(X_1)|f_2(X_2), U) = 0$, we have $H(f_1(X_1)|U) = 0$. This concludes that there exist a deterministic function $f$ such that $f(U) = f_1(X_1) = f_2(X_2)$.

Next, we prove the other direction (a) $\implies$ (b). Note that $W$ given in the proposition statement satisfies $W = g_1(X_1) = g_2(X_2)$ and therefore, from (a), we know that there exist a function $h_1$ such that $W = h_1(U)$. Since $W$ also satisfies that $X_1 \perp\!\!\!\perp X_2 \mid W$ and $U = s_1(X_1) = s_2(X_2)$, then applying the direction (b) $\implies$ (a), we have that $U = h_2(W)$ for some function $h_2$. Therefore, there is a one-to-one mapping between $W$ and $U$. □

Note that it is difficult to obtain a random variable $U$ that satisfies (a) (i.e. the sufficient common variable in Defined 2.2). The Proposition A.1 allows us to find a random variable $W$ (if it exists) instead. And the one with minimum entropy can be obtained by solving the optimization problem (2).

### A.2 Derivation of the Shared Loss (4)

We first group the terms that only depend on $U_1$ or $U_2$ as follows.

$$\mathcal{L}_{\text{info}}^{\text{shared}} = \alpha d(U_1, U_2) + \beta(H(U_1) + H(U_2)) + I(X_1; X_2 \mid U_1) \quad (13)$$
$$+ I(X_1; X_2 \mid U_2)$$
$$= \alpha d(U_1, U_2) + \mathcal{L}(U_1) + \mathcal{L}(U_2), \quad (14)$$

where $d(U_1, U_2)$ can be measured using the Euclidean distance or other distance measures. And

$$\mathcal{L}(U_1) = I(X_1; X_2|U_1) + \beta H(U_1)$$
$$\overset{(i)}{=} I(X_1; X_2|U_1) + \beta I(X_1; U_1)$$
$$\overset{(ii)}{=} \mathbb{E}_{U_1}\left[D_{KL}(p_{X_1,X_2|U_1}||p_{X_1|U_1}p_{X_2|U_1})\right] \quad (15)$$
$$+ \beta D_{KL}(p_{X_1,U_1}||p_{X_1}p_{U_1})$$
$$= \mathbb{E}_{X_1,X_2,U_1}\left[\log \frac{p_{X_1,X_2|U_1}}{p_{X_1|U_1}p_{X_2|U_1}}\right] + \beta\mathbb{E}_{X_1,U_1}\left[\log \frac{p_{X_1,U_1}}{p_{X_1}p_{U_1}}\right]$$
$$= \mathbb{E}_{X_1,X_2,U_1}\left[\log \frac{p_{X_1,X_2,U_1}p_{U_1}}{p_{X_1,U_1}p_{X_2,U_1}}\right] + \beta\mathbb{E}_{X_1,U_1}\left[\log \frac{p_{X_1,U_1}}{p_{X_1}p_{U_1}}\right]$$
$$= \mathbb{E}_{X_1,X_2,U_1}\left[\log \frac{p_{X_1,X_2,U_1}}{p_{X_1}p_{X_2}p_{U_1}} + \log \frac{p_{X_1}p_{U_1}}{p_{X_1,U_1}} + \log \frac{p_{X_2}p_{U_1}}{p_{X_2,U_1}}\right] \quad (16)$$
$$+ \beta\mathbb{E}_{X_1,U_1}\left[\log \frac{p_{X_1,U_1}}{p_{X_1}p_{U_1}}\right]$$
$$= \mathbb{E}_{X_1,X_2,U_1}\left[\log \frac{p_{X_1,X_2,U_1}}{p_{X_1}p_{X_2}p_{U_1}}\right. \quad (17)$$
$$\left. +(1 - \beta)\log \frac{p_{X_1,U_1}}{p_{X_1}p_{U_1}} + \log \frac{p_{X_2,U_1}}{p_{X_2}p_{U_1}}\right], \quad (18)$$

where $(i)$ follows the relation between mutual information an entropy that $I(X_1; U_1) = H(U_1) - H(U_1|X_1)$ and $H(U_1|X_1) = 0$ because $U_1$ is a deterministic function of $X_1$; $(ii)$ is by definition of the conditional mutual information; and the remaining equalities use the Bayes' rule. Similarly, we have

$$\mathcal{L}(U_2) = I(X_1; X_2|U_2) + \beta H(U_2)$$
$$= \mathbb{E}_{X_1,X_2,U_2}\left[\log \frac{p_{X_1,X_2,U_2}}{p_{X_1}p_{X_2}p_{U_2}}\right. \quad (19)$$
$$\left. +\log \frac{p_{X_1,U_2}}{p_{X_1}p_{U_2}} + (1 - \beta)\log \frac{p_{X_2,U_2}}{p_{X_2}p_{U_2}}\right]. \quad (20)$$

Combining (14), (18) and (20), we can obtain

$$\mathcal{L}_{\text{info}}^{\text{shared}} = \alpha d(U_1, U_2) + \sum_{i=1}^{2}\mathbb{E}_{X_1,X_2,U_i}\left[\log \frac{p_{X_1,X_2,U_i}}{p_{X_1}p_{X_2}p_{U_i}}\right. \quad (21)$$
$$\left. +(1 - \beta)\log \frac{p_{X_i,U_i}}{p_{X_i}p_{U_i}} + \log \frac{p_{X_{3-i},U_i}}{p_{X_{3-i}}p_{U_i}}\right]. \quad (22)$$

### A.3 Derivation of the Private Loss (7)

Similar to (18), since $H(V_1|X_1) = H(V_2|X_2) = 0$, we have that

$$\mathcal{L}_{\text{info}}^{\text{private}} = \gamma H(V_1) + \gamma H(V_2) + \epsilon I(V_1; U_1) + \epsilon I(V_2; U_2),$$
$$= \gamma I(X_1; V_1) + \gamma I(X_2; V_2) + \epsilon I(V_1; U_1) + \epsilon I(V_2; U_2), \quad (23)$$
$$= \sum_i \mathbb{E}_{X_i,V_i,U_i}\left[\gamma \log \frac{p_{X_i,V_i}}{p_{X_i}p_{V_i}} + \epsilon \log \frac{p_{V_i,U_i}}{p_{V_i}p_{U_i}}\right].$$

**Table 7: Statistical Summaries of Individual Domains**

| Dataset | Applications | Classes | Modality Sampling Rate | Sample Length | Interval (Overlap) | #Pretrain Samples | Alignment | #Alignment Samples | # Finetune Samples |
|---------|-------------|---------|----------------------|---------------|--------------------|-------------------|-----------|-------------------|--------------------|
| Domain M | MOD | 7 | acoustic (8kHz), seismic (100Hz) | 2 sec | 0.2 sec (0%) | 39,609 | ✓ | 1981 | 734 |
| Domain G | MOD | 4 | acoustic (8kHz), seismic (100Hz) | 2 sec | 0.2 sec (0%) | 35,168 | ✗ | - | - |
| Domain T | MOD | 4 | acoustic (8kHz), seismic (100Hz) | 2 sec | 0.2 sec (0%) | 43,819 | ✗ | - | - |
| PAMAP2 | HAR | 18 | acc, gyro, mag (all 100Hz) | 2 sec | 0.4 sec (50%) | 9,611 | ✓ | 4805 | 961 |
| RealWorld-HAR | HAR | 8 | acc, gyro, mag (all 50Hz) | 5 sec | 1 sec (50%) | 12,887 | ✗ | - | - |

**Table 8: Statistical Summaries of Evaluated Datasets.**

| Dataset | Classes | Modalities (Freq) | Sample Length | Interval (Overlap) | #Samples | #Labels |
|---------|---------|------------------|---------------|--------------------|-----------|---------|
| Domain M | 7 | acoustic (8kHz), seismic (100Hz) | 2 sec | 0.2 sec (0%) | 39,609 | 7,335 (M); 3,136 (G); 4,205 (T) |
| ACIDS | 9 | acoustic, seismic (both 1025Hz) | 1 sec | 0.25 sec (50%) | 27,597 | 27,597 |
| RealWorld-HAR | 8 | acc, gyro, mag, lig (all 50Hz) | 5 sec | 1 sec (50%) | 12,887 | 12,887 |
| PAMAP2 | 18 | acc, gyr, mag (all 100Hz) | 2 sec | 0.4 sec (50%) | 9,611 | 9,611 |

# B  Datasets

This section describes the cross-modal alignment and joint multi-modal pretraining evaluation datasets. We have two real-world IoT applications: Moving Object Detection (MOD) and Human Activity Recognition (HAR).

## B.1  Cross-modal Alignment Datasets

*B.1.1  Moving Object Detection.* We have seismic and acoustic signals describing different vehicles on three different domains. For simplicity, we use one letter to represent each domain. Table 7 provides the statistical values of each domain, and we provide detailed descriptions below:

**Domain M** is a publicly released [40] moving object detection dataset consisting of seismic and acoustic data from 7 different moving vehicles, recorded at three different distances and four different speeds. The sensor nodes, RaspberryShake, include a microphone array sampled at 16,000Hz and a geophone sampled at 100Hz. This dataset contains three types of downstream tasks — vehicle classification, distance classification, and speed classification. For cross-validation, data collected from three sensor nodes are used for training, and data from a separate node is used for testing.

**Domain G** contains a self-collected dataset on state park grounds near an outdoor research facility. Four sensor nodes, each featuring a geophone and a microphone array, were deployed to collect seismic and acoustic vibration signals from nearby objects at 200Hz and 16000Hz, respectively. Both seismic and acoustic signals were downsampled by half to match the signals from MOD. Four targets were chosen during the data collection, and each navigated the neighborhood near the sensors in some arbitrary order. Targets involved were (1) Polaris[3] off-road vehicle, (ii) a Warthog[4] all-terrain unmanned ground robot, (iii) a Husky unmanned outdoor field robot[5], and (iv) a standard civilian truck. Data collection spanned over two days with different on-site noises observed, including but

not limited to human interference (talking and walking near the sensors), environmental disturbances, background noises, etc.

**Domain T** contains seismic and acoustic vibration signals with a similar setup as MOD but involves different targets and scenes. This set contains data collected from a paved parking lot, unpaved trails, and gravel roads within a park. Vibration signals of 2 standard-size SUVs from different manufacturers, one lightweight sports car, and one muscle car were recorded. For each scene, we collected one hour of data for each vehicle. We use the first 50 minutes for training and the last 10 minutes for validation and testing.

*B.1.2  Human Activity Recognition.* Unlike the MOD application, where we used data from different domains for unimodal pretraining, we leveraged two different HAR datasets for unimodal pretraining and cross-modal alignment to evaluate the scenario in which IMU data has high degrees of heterogeneity.

**RealWorld-HAR [62]** is a public dataset that utilizes accelerometer, gyroscope, magnetometer, and light signals sampled at 50Hz. It includes data from 15 subjects performing eight common human activities: climbing stairs down and up, jumping, lying, standing, sitting, running/jogging, and walking. We used the data collected from devices positioned at the subjects' waists. For our experiments, we randomly selected ten subjects for training, 2 for validation, and 3 for testing.

**PAMAP2 [57]** contains inertial data from 18 human daily activities, such as walking, cycling, and playing soccer, performed by nine subjects. The dataset includes 9,611 instances, with data captured using inertial measurement units (IMUs) placed on the chest, the wrist of the dominant arm, and the dominant side's ankle. However, our experiment only utilized data collected from the wrist. Each data contains a 3-axis accelerometer, gyroscope, and magnetometer signal at a sampling rate of 100Hz. Seven random subjects are used for training, and two subjects for testing.

**Combined** is a concatenated dataset of RealWord-HAR and PAMAP2. Since PAMAP2 does not contain any light signals, we drop the light modality and only use the accelerometer, gyroscope, and magnetometer for evaluation.

---

[3]https://www.polaris.com/
[4]https://clearpathrobotics.com/warthog-unmanned-ground-vehicle-robot/
[5]https://clearpathrobotics.com/husky-unmanned-ground-vehicle-robot/

**Table 9: Encoder & Decoder configurations.**

| Dataset | MOD | ACIDS | RealWorld-HAR | PAMAP2 |
|---|---|---|---|---|
| Dropout Ratio | 0.2 | 0.2 | 0.2 | 0.2 |
| Patch Size | aud: [1, 40], sei: [1,1] | [1, 8] | [1, 2] | [1, 2] |
| Window Size | [3, 3] | [2,4] | [3, 3] | [3, 5] |
| Encoder Block Num | [2, 2, 4] | [2, 2, 4] | [2, 2, 2] | [2, 2, 2] |
| Encoder Block Channels | [64, 128, 256] | [64, 128, 256] | [32, 64, 128] | [32, 64, 128] |
| Head Num | 4 | 4 | 4 | 4 |
| Encoder Fusion Channel | 256 | 256 | 128 | 128 |
| Encoder Fusion Head Num | 4 | 4 | 4 | 4 |
| Encoder Fusion Block | 2 | 2 | 2 | 2 |
| FC Dim | 512 | 512 | 256 | 128 |
| Factorization Dimension | 128 | 128 | 128 | 128 |
| Decoder Block Num | [2, 2] | [2, 2] | [2, 2] | [2, 2] |
| Decoder Block Channels | [128, 64] | [128, 64] | [64, 32] | [64, 32] |

## B.2   Joint Multimodal Pretraining

In addition to pair-efficient SSL, we separately evaluate standard multimodal SSL, pretraining on large-scale multimodal data. We describe each dataset in more detail below and summarize the statistics in Table 8

**Moving Object Detection (MOD) [40]** is the superset of the domains we used in cross-modal alignment evaluation. We pretrain the encoders on domain M and then report the finetune performance in domain G and domain T.

**Acoustic-Seismic Identification Data Set (ACIDS)** is an additional multimodal dataset collected using two synchronized acoustic and seismic sensor systems (16-bit analog-to-digital converter operating at 1025 Hz) with more than 270 individual data runs, each featuring a singular ground vehicle type. The dataset captures signals from 9 distinct ground vehicles operating under 3 different environmental conditions. These targets were recorded at a constant speed from 5km/h to 40km/h as they navigated through the sensor systems, passing the sensors between 25 and 100 meters. The acoustic data is processed with a low-pass filtration at 400 Hz using a 6th-order filter to obviate spectral aliasing and a high-pass filtration at 25 Hz with a 1st-order filter to mitigate environmental noise (*e.g.*, wind). The collected data runs were randomly divided into an 8:1:1 ratio for training, validation, and testing.

**RealWorld-HAR [62]** is the same dataset we used for cross-modal alignment evaluation. See B.1.2 for the detailed description.

**PAMAP2 [57]** is the same dataset we used for cross-modal alignment evaluation. See B.1.2 for the detailed description.

## C   Data Preprocessing

We partition the time-series data into segments of uniform length. Each segment is then subdivided into intervals that may or may not overlap. We apply the Fourier transform to the signal in each interval to derive its spectral content, thereby retaining both temporal and spectral characteristics. The resultant spectrograms are subsequently inputted into our designed feature encoders. We have established a suite of data augmentation techniques applicable to the time domain pre-Fourier transform and the frequency domain post-Fourier transform. Each sample is subjected to a randomly chosen augmentation, which could be from either domain. Moreover, to enhance the stochastic nature of data augmentation within multimodal frameworks, we assign a fifty percent chance for each modality to undergo the randomly selected augmentation process.

## D   Backbone

**SWIN-Transformer[41].** SWIN-Transformer is a variant of the Vision Transformer [14] designed for images. We have adapted the SWIN-Transformer to process time-frequency spectrogram inputs. The time-frequency spectrogram input from each modality is first segmented into non-overlapping patches of embedding vectors via a convolutional layer. The model then extracts features through blocks of layer, each consisting of self-attention layers computed within the local windows. A shifting window mechanism is applied to increase the perceptual field at a much lower computational cost and allows the model to capture global information. The patches are downsampled at the end of each block by merging adjacent patches to double the feature channels. Separate SWIN-Transformer encoders are used to extract features from each sensory input modality. For supervised learning, additional self-attention layers are used to fuse the features to combine information across the various modalities. In cases where the learning framework operates at the modality level, the model bypasses the cross-modal fusion stages and computes pretraining losses directly on the features extracted from each modality. SWIN-Transformer is also used as the decoder for Masked Autoencoders. Instead of downsampling after each block, we expand the patches through linear layers at the beginning of each block to mirror the encoding steps. The decoded features at the end are then projected to match the dimension of patched input spectrograms for reconstruction loss. The backbone configurations of the datasets used in this paper for both encoding and decoding are detailed in Table 9.

## E   Experiment and Implementation Details

**Training**. We specify the configurations used for InfoMAE during Joint Multimodal Pretraining, InfoMAE two-stage pretraining, and finetuning in Table 10. We randomly sample a batch of sequences of 4 consecutive samples (a total of 256 samples) during

**Table 10: Training configurations.**

|  | Joint Multimodal Pretraining | Unimodal Pretraining | Cross-Modal Alignment | Finetuning |
|---|---|---|---|---|
| Optimizer | AdamW [43] | AdamW [43] | AdamW [43] | Adam [32] |
| Weight Decay | 0.05 | 0.05 | 0.05 | 0.05 |
| Start Learning Rate (LR) | 0.0001 | 0.0001 | 0.0001 | 0.01 |
| LR Scheduler | Cosine | Cosine | Cosine | Step |
| LR Decay | 0.2 | 0.2 | 0.2 | 0.2 |
| LR Period | 500 | 500 | 100 | 50 |
| Epochs | MOD, ACIDS: 2500 RealWorld-HAR, PAMAP2: 1000 | 2500 | 500 | 200 |
| Batch Size | 256 | 256 | 256 | 128 |

**Table 11: Discriminator configurations.**

| Dataset | MOD | ACIDS | RealWorld-HAR | PAMAP2 |
|---|---|---|---|---|
| Dropout Ratio | 0.2 | 0.2 | 0.2 | 0.2 |
| Mod Conv Kernel | aud: [1, 5], sei: [1,3] | [1,4] | [1, 3] | [1, 5] |
| Mod Conv Channel | 128 | 128 | 128 | 64 |
| Mod Conv Layers | 5 | 6 | 6 | 4 |
| MLP Layers | 4 | 4 | 4 | 4 |
| Activation Function | LeakyReLU | LeakyReLU | LeakyReLU | LeakyReLU |

**Table 12: Discriminator training configurations.**

| Stage | Joint Multimodal Pretraining | Cross-Modal Alignment |
|---|---|---|
| Optimizer | AdamW [43] | AdamW [43] |
| Weight Decay | 0.005 | 0.005 |
| Start LR | 0.00005 | 0.00005 |
| LR Scheduler | Cosine [42] | Cosine [42] |
| LR Decay Epochs | MOD, ACIDS: 500 RealWorld-HAR, PAMAP2: 50 | MOD, ACIDS: 500 RealWorld-HAR, PAMAP2: 50 |
| Warm Up Epochs | 10 | 10 |
| Train Period | 50 | 50 |

pretraining. We jointly optimize the backbone encoders and decoders with AdamW [43] optimizer and Cosine scheduler [42]. The model configurations for both the encoder and decoder are summarized in Table 9. Additionally, when training InfoMAE, we jointly train discriminators for density-ratio estimations [31, 50, 60]. We apply convolution blocks to map the time-frequency sample into a one-dimensional embedding to match the input dimension $X_1$ with their shared and private representations $V_1, U_1$. Then, the discriminator estimates their density ratio through a 5-layer MLP. The exact discriminator configuration is provided in Table 11. We follow standard practice [31] in training the discriminators with warmup epochs and training periods. The detailed training configurations for the discriminators are presented in Table 12.

**Computation**. We conducted our experiments on NVIDIA RTX 4090 GPUs with 24GB memory. The training time varies from a few minutes for finetuning to 2 days for Joint Multimodal Pretraining.

**Implementations**. We build our models on top of the open-source implementations [24, 40, 41] using Pytorch 2.0.1. We will also release our code upon acceptance.

# F  Baselines
We describe all the baselines used in the evaluation below.

**Supervised:** The supervised approach trains with full supervision for each task to learn a mapping from the input to task-specific labels. Labels are used to update the entire model which includes both the encoder and the classification layer.

**SimCLR** [6] presents a simple contrastive framework for contrastive learning in visual perception tasks. In our implementation, each batch is randomly sampled. During the pretraining phase, we apply random augmentations to each sample to create two distinct views. The goal of these augmentations is to pull the transformed version of the same sample closer in the feature space while pushing the representations of other samples further apart. We treat other samples within the same minibatch (2N - 2) as the negative pairs. Consequently, variant views of an identical sample form positive pairs, whereas views derived from separate samples are treated as negative pairs.

**MoCoV3** [8] involves a query encoder denoted as $f_q$ and a key momentum encoder denoted as $f_k$. Both encoders have the same backbone network followed by a projection head. The query encoder $f_q$ includes an additional projection head at the end. During pretraining, MoCoV3 relies on randomly augmented views of input samples to learn transformation invariant features. For each sample, it generates a query vector $q$ using $f_q$ and a key vector $k$ using $f_k$. The objective is to maximize the agreement between positive encoded query-key pairs. Specifically, the positive key $k^+$ is encoded from the same sample as the query $q$, while the negative keys $k^-$ are encoded from other samples within the same mini-batch. This encourages the model to learn meaningful representations by contrasting positive and negative pairs. Additionally, MoCoV3 employs random batch sampling during training, and the key momentum encoder $f_k$ is gradually updated using a query momentum with the query encoder $f_q$.

**CMC** [63] represents a novel approach in contrastive learning that emphasizes the utilization of multiview data. CMC extracts meaningful representations by treating different modalities of the

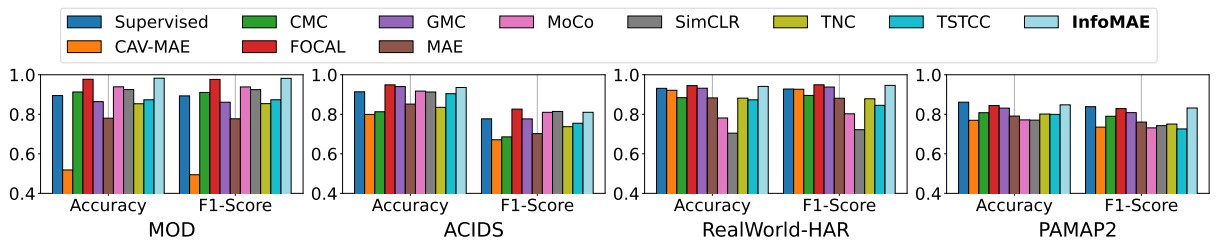

**Figure 6: Joint Multimodal Pretraining compared with previous joint pretraining SSL frameworks on four datasets.**

data as different views, minimizing the geometric gap between synchronized feature representations from these different modalities. During the training process, each mini-batch is processed by the backbone model to obtain modality representation. The framework then focuses on maximizing the similarity of representations from identical samples across modalities, while simultaneously treating dissimilar modality representations from distinct samples as negative pairs. The cumulative loss from all modality pairings is computed as the final loss term.

**GMC** [55] presents a multimodal contrastive loss function designed to align embeddings from various modalities geometrically to the joint embeddings. This framework operates on the principle of random batch sampling and augmentation similar to other contrastive learning baselines. In addition to modality-specific encoders, GMC has an additional joint encoder that processes inputs from all modalities concurrently. To facilitate the alignment of the joint embedding with the individual modality embeddings, an extra linear layer maps the joint embedding into the same dimensional space. Subsequently, a unified projection head is utilized to project both the individual and joint modality embeddings before the loss computation. GMC aims to maximize the similarity of individual modality embeddings (single view) to the joint embedding (global view).

**FOCAL** [40] is a recent contrastive learning framework designed for multimodal time-series signals. FOCAL acknowledges that each modality contains information shared across different modalities and information unique to each modality. FOCAL follows CMC in learning the shared representations and SimCLR for private representation. To avoid entanglement of the factorized representations, FOCAL minimizes the cosine similarity between the shared and private subspaces as well as between the private subspaces, creating an orthogonal latent space. In addition, FOCAL proposes a temporal ranking constraint to learn time-series locality.

**TNC** [64] is a self-supervised learning framework that captures time series representations through a debiased contrastive objective, distinguishing between temporally close and distant samples. TNC defines neighboring samples as those within the same sequence, sharing similar timestamps, and non-neighboring samples as those from different sequences. A discriminator is employed to predict the likelihood of each sample and its neighbors being in the same temporal window, to maximize the similarity of neighboring samples while minimizing that of non-neighboring ones.

**TS-TCC** [16] is a contrastive learning framework that robustly captures time series representations. It achieves this by combining cross-view predictions and contrasting both temporal and contextual information. In practice, TS-TCC randomly groups multiple sequences into mini-batches. For each sample, it generates two views through random augmentations. Context vectors are extracted from all sample representations up to a given timestamp within the sequence using an autoregressive model. These context vectors are then used to predict future timestamps in the other view. The framework simultaneously addresses both temporal alignment and context awareness, enhancing its ability to discern meaningful patterns in time series data.

**MMAE** [24] is a variant of Masked Autoencoders (MAE) with additional fusion modules for multimodal learning. It incorporates an encoder-decoder architecture and achieves SOTA performance on multiple vision tasks. Unlike contrastive learning, MAE does not depend heavily on random augmentations. During the pretraining, we randomly mask a significant portion (*i.e.*, 75%) of each modality input. Instead of dropping the masked patches as in the original MAE paper, we replace them with 0 values to ensure consistent dimensions. A separate encoder and decoder are used for each modality. After encoding, we concatenate the modality features and then use separate MLP projection layers to get the fused modality embeddings before decoding to learn cross-modal information. Then a separate projector is used to map the fused embedding back to each modality embedding. Finally, the modality decoder reconstructs the modality input from the projected modality embeddings. The overall objective is to minimize the mean squared error (MSE) between the masked portion of the original modality patches and the reconstructed modality patches.

**CAV-MAE** [18] is a self-supervised learning framework building on top of both contrastive learning and MAE to learn audio-visual representations. It extends the capabilities of the traditional Masked Autoencoder (MAE) to process and learn from both audio and visual inputs simultaneously. CAV-MAE first extracts modality embedding through individual modality encoder. Then, it applies a joint encoder to extract joint embedding as well as encode separate modality embedding. The joint embedding is decoded for reconstruction, while the encoded modality embeddings are treated as positive pairs for contrastiyve learning.

# G  Additional Evaluation

## G.1  Joint Multimodal Pretraining

Although InfoMAE is primarily designed for learning settings where the multimodal pairs are scarce, InfoMAE also demonstrates strong flexibility and generalization as a standard multimodal SSL framework when abundant multimodal pairs are available. We present

**Table 13: Ablation Results of InfoMAE.**

| Dataset | MOD | | ACIDS | | RealWorld-HAR | | PAMAP2 | |
|---|---|---|---|---|---|---|---|---|
| Frameworks | Acc | F1 | Acc | F1 | Acc | F1 | Acc | F1 |
| woTemp | 0.8734 | 0.8724 | 0.8808 | 0.7154 | 0.8442 | 0.8394 | 0.6948 | 0.6279 |
| woShared | 0.9531 | 0.9518 | 0.8845 | 0.7435 | 0.8771 | 0.8843 | 0.8095 | 0.7686 |
| woPrivate | 0.9082 | 0.9066 | 0.8562 | 0.7174 | 0.9100 | 0.9179 | 0.8080 | 0.7680 |
| woAugmentation | 0.9538 | 0.9532 | 0.9101 | 0.7275 | 0.9106 | 0.9180 | 0.8163 | 0.7903 |
| InfoMAE | 0.9826 | 0.9819 | 0.9356 | 0.8101 | 0.9411 | 0.9462 | 0.8478 | 0.8319 |

additional finetuning performance after joint multimodal pretraining in Figure 6 with many SOTA multimodal SSL frameworks across the four real-world IoT datasets we described. InfoMAE significantly exceeds the MAE-based SSL framework and achieves comparable or superior performance to the contrastive learning baselines. It is noteworthy that other baselines are mainly designed for joint multimodal pretraining. InfoMAE is a universal framework for cross-modal alignment that achieves comparable performance as multimodal SSL with few sacrifices.

## G.2 Ablation Studies on Joint Multimodal Pretraining

In addition to ablating InfoMAE for cross-alignment, we conduct ablation studies on Joint Multimodal Pretraining, evaluating different variants of InfoMAE when abundant multimodal data is available for pretraining. The results across the four datasets are presented in Table 13. Consistent with the results in Section 4.6, removing either shared or private representations hurts the performance of InfoMAE, indicating that both modality-shared and modality-exclusive information contribute positively to downstream tasks. Removing augmentations has minor impacts. However, InfoMAE experiences the most significant degradation when the temporal locality is absent, primarily due to that a generalized learning objective is required for Information Formulation to learn meaningful factorizations on time-series signals.

