# OpenReview forum: "InfoMAE: Pairing-Efficient Cross-Modal Alignment with Informational Masked Autoencoders for IoT Signals"
_ACM.org/TheWebConf/2025/Conference — WWW 2025 Poster_

### Official Review · Reviewer_uQgK · 2024-12-01

**Novelty:** 6
**Technical Quality:** 5

**Review:**

This paper presents an innovative self-supervised learning framework, InfoMAE, for joint multimodal pretraining, with a focus on real-world IoT applications like Moving Object Detection (MOD) and Human Activity Recognition (HAR). The proposed framework addresses the challenges associated with multimodal data, particularly in terms of aligning and pretraining cross-modal features when multimodal pairs are scarce. It is evaluated across multiple datasets, including MOD, ACIDS, RealWorld-HAR, and PAMAP2, and shows promising performance compared to state-of-the-art contrastive learning frameworks.

## Strengths:
1. The work introduces a novel approach that integrates aspects of both contrastive learning and Masked Autoencoders (MAE) into a unified framework, InfoMAE. The hybrid approach shows promise in improving multimodal alignment, especially in scenarios where data pairs are limited.

2. The authors clearly identify the need for an effective multimodal SSL framework that performs well even when multimodal pairs are scarce, which is an important and relevant research direction in AI.

3. The paper provides extensive experiments on multiple datasets, demonstrating the flexibility and robustness of InfoMAE across various real-world IoT applications. The performance metrics (accuracy and F1 score) are convincingly presented and show that InfoMAE outperforms previous approaches.

4. The choice of datasets is well-justified, and the use of multiple evaluation metrics and ablation studies adds depth to the analysis. The authors have clearly demonstrated the efficacy of the proposed method.

5. The framework has significant potential for practical applications, especially in scenarios where multimodal data is available but alignment and pretraining methods are still in need of improvement.

## Weaknesses:
1. Some parts of the methodology could benefit from clearer explanations, particularly regarding how the fusion of modality-specific features is achieved in InfoMAE. While the method is described in detail, it may be challenging for readers to fully understand the subtleties of the joint encoder-decoder architecture and the use of both shared and private representations. Simplifying these explanations would increase the accessibility of the paper.

2. The paper does not sufficiently discuss the limitations of the proposed approach. For example, the model's performance under extreme data sparsity (i.e., when multimodal pairs are exceedingly rare) could be further explored, and the trade-offs between computational complexity and model performance are not fully addressed. An analysis of these factors could enhance the completeness of the paper.

3. While the authors compare InfoMAE with several multimodal SSL baselines, there is limited discussion of alternative joint multimodal pretraining methods that do not rely on contrastive learning or MAE-like frameworks. Expanding this comparison to include other methods could provide more context and highlight InfoMAE’s relative advantages.

4. The role of data augmentation in multimodal pretraining is mentioned, but a more detailed analysis of its impact on model performance would be valuable. The authors briefly state that augmentations have minor impacts, but a deeper exploration of how different augmentation strategies affect the results could lead to interesting insights.

5. The paper could benefit from a discussion on the computational cost and scalability of InfoMAE, particularly given that the model requires substantial computational resources for training (e.g., 2 days for Joint Multimodal Pretraining on an NVIDIA RTX 4090). A more thorough discussion of the trade-off between model complexity and performance would be useful for practitioners considering the deployment of the framework.

## Suggestions for Improvement:
- Clarify the explanation of the encoder-decoder architecture, particularly the fusion of shared and private representations.
- Include a more thorough discussion of the model's limitations and potential weaknesses, especially in terms of computational cost and data sparsity.
- Provide more detailed comparisons with other joint multimodal pretraining approaches.
- Expand the analysis of the role of data augmentation in multimodal learning, particularly its effects on performance.
- Discuss the potential scalability of the framework in real-world applications and its suitability for low-resource settings.

**Questions:**

1. Can the authors provide more details on how the fusion of modality-specific features is handled in the encoder-decoder architecture? Specifically, how are the shared and private representations integrated during the pretraining phase, and what is the impact of this fusion on the overall model performance?

2. Could the authors elaborate on the potential limitations of InfoMAE in scenarios where multimodal pairs are extremely rare? Are there any conditions under which the framework might fail to perform adequately?

3. How does InfoMAE compare to alternative joint multimodal pretraining methods that do not rely on contrastive learning or MAE frameworks? Could the authors provide additional insights into how these methods would perform on the datasets used in this study?

4. What specific data augmentation techniques were applied to the time-domain and frequency-domain signals, and how do they influence the model’s ability to learn meaningful features? Could the authors explore the impact of different augmentation strategies in more detail?

5. Given the substantial computational resources required for training, how does InfoMAE scale to larger datasets or more complex real-world applications? Are there any optimizations that could be made to reduce the computational cost of training?

6. Could the authors discuss the trade-offs between model complexity and performance, particularly with respect to the use of multiple modalities and the encoder-decoder architecture?

7. In terms of practical applications, how does InfoMAE perform when deployed in environments with noisy or sparse sensor data? Are there any strategies for improving performance in these challenging settings?

8. What are the potential real-world applications of InfoMAE beyond the datasets and domains considered in this paper? Are there specific IoT scenarios where this framework would be particularly beneficial?

**Reviewer Confidence:**

4: The reviewer is certain that the evaluation is correct and very familiar with the relevant literature

**Scope:**

4: The work is relevant to the Web and to the track, and is of broad interest to the community

---

### Official Review · Reviewer_ebqq · 2024-12-01

**Novelty:** 5
**Technical Quality:** 5

**Review:**

This work tackles the challenge of multimodal pair efficiency in a self-supervised learning setup by introducing a cross-modal alignment framework, InfoMAE.

The key highlight of the proposed solution is that it could achieve efficient cross-modal alignment with limited data pairs by leveraging information theory concepts that address both instance-level and distribution-level alignment.

The experimental results on two different datasets (moving object detection and human activity recognition) show a clear advantage of using the proposed solution over the state-of-the-art literature.

My main concern is that the paper does not seem to fit the scope of the conference or the track. The contribution is more focused on bringing innovation to learning architecture. The same solution could be applicable to address the general challenges of handling multimodality in a given dataset under SSL setup. I do not find any web-related component in this work. Details regarding system-level or algorithmic optimization techniques (if any) for managing multimodal streaming data that are missing.

There are a few typos like floating dots (line no. 432), quotations (line no 491) and symbolic ambiguity (like S (spectrum length) and s in the definition 2.1 (not defined, not clear) in the paper.

The paper's content includes too much redundancy in my opinion. The introduction section can be made more concise, the last part of the first part of the Introduction section seems repetitive.

The problem definition includes part of the methodology which has already been introduced once in the Introduction and then again in section 3.

**Questions:**

What is meant by "...it remains unclear whether this is sufficient when only limited modality pairs are available for learning". Why is it so? Is there any particular reason that this is still unaddressed, or any relevant citation to claim this?

How feasible is the assumption that samples across the modalities are synchronized? --Line no. 197.

I wonder how the selection of window size would impact the solution pipeline since it will be a crucial deciding factor for the quality of the alignment. Is there any empirical observation on the optimal window size or will it have any implication on the outcome?

What is the scale of labeled data for supervised learning and fine-tuning? Is it purely a data-driven selection?

Line 218 and 253, the relation between |Xi| and |Xi^s|, aren't they contradictory?

What does this M,G,T represent (ref datasets)?

Do you intend to reduce the amount of multimodal data? or the number of modalities itself? (ref line no. 795)

**Reviewer Confidence:**

2: The reviewer is willing to defend the evaluation, but it is likely that the reviewer did not understand parts of the paper

**Scope:**

1: The work is irrelevant to the Web

---

### Official Review · Reviewer_JQ9p · 2024-12-03

**Novelty:** 3
**Technical Quality:** 3

**Review:**

Pros:
The focus on multi-modal issues within IoT research is engaging and tackles a relevant challenge in the field.
The visual explanations of the proposed techniques are well-designed and help in understanding the methodology.
While the reasons for using specific techniques require more detail, the overall presentation is clear and accessible.

Cons:
The problem definition lacks clarity compared to prior studies, making it challenging to assess the novelty of the work.
The connection between the proposed problem and the solution is not sufficiently justified, requiring a clearer rationale.
The paper needs to better articulate its contributions to enhance the overall impact and make the key advancements explicit.

**Questions:**

1) How is the proposed work connected to web technologies? Clarifying this link would improve the contextual relevance.

2) What does "multi-modal" specifically refer to in this paper? Does it solely pertain to multi-modal sensors, or does it extend to modalities like vision and other domains? A more precise definition would be helpful.

3) Can you provide a concrete, data-driven example in the first paragraph of the Introduction to strengthen the motivation and make the problem relatable?

4) What are the specific limitations of existing multi-modal SSL frameworks that your work addresses? A clearer explanation beyond Figure 1 is necessary to understand the shortcomings of previous approaches.

5) The statement "we observe that with limited multimodal pairs, we can effectively convert independently trained unimodal encoders into a coherent model that sustains strong generalizability in multimodal tasks" is intriguing. What evidence or experimental results support this claim?

6) Do the sensors used in your experiments adequately capture the characteristics of the challenges outlined in the problem statement? Addressing this would enhance the experimental validity.

7) The overall quality of the paper needs refinement, as there are instances of punctuation errors (e.g., "",.") and minor formatting issues. Improving these details would enhance readability and professionalism.

8) An overarching guideline for why each experiment is conducted and how they connect to the main contributions should be added to improve clarity and structure.

**Reviewer Confidence:**

3: The reviewer is confident but not certain that the evaluation is correct

**Scope:**

1: The work is irrelevant to the Web

---

### Official Review · Reviewer_D3Lv · 2024-12-03

**Novelty:** 4
**Technical Quality:** 5

**Review:**

This paper presents InfoMAE, a novel framework designed to address the challenges of cross-modal alignment in IoT signals, particularly under conditions where multimodal pairings are sparse. InfoMAE leverages an information-theoretic approach to enforce both distribution-level and instance-level alignment, facilitating the integration of pretrained unimodal representations into cohesive multimodal models.

Strength
1. The theoretical derivation in this paper aligns well with the distribution-level alignment described in the text. Combined with the experimental results, the proposed method of achieving semantic alignment through representation factorization is well-founded.
2. The presentation of this paper is very clear and easy to follow.
3. The experiments in this paper demonstrate significant performance improvements.

Weakness
1. The optimization objective defined in Section 2.2 and the implementation details in Section 3.2 are not well described in terms of their equivalence. It is recommended to add more content to explain how these two align.
2. Although the proposed method is well-founded, it appears to be an integration of existing techniques. Both representation factorization and information-theoretic distribution-level alignment are established approaches. Compared to prior works, the reviewer believes that the main contribution of this paper lies in applying distribution-level alignment to a specific subdomain.

**Questions:**

1. Under sparse sampling conditions, even distribution-based alignment is prone to sampling errors, as distributions are estimated based on observed samples. Furthermore, the paper employs numerous approximations and custom coefficients, which might lead to suboptimal performance compared to the expectations set by the theoretical framework.
2. While the paper highlights "Pairing-Efficient" in both the title and abstract, the experimental results primarily focus on accuracy comparisons. The reviewer suggests including an analysis of how the performance gap between InfoMAE and the baselines changes with varying sample sizes

**Reviewer Confidence:**

3: The reviewer is confident but not certain that the evaluation is correct

**Scope:**

3: The work is somewhat relevant to the Web and to the track, and is of narrow interest to a sub-community